# Christian Women and the Development of Nascent Feminist Consciousness in Nineteenth-Century China

**Anneke H. Stasson**

John Wesley Honors College, Indiana Wesleyan University, Marion, IN 46953, USA;
anneke.stasson@indwes.edu

**Abstract:** In 2010, Padma Anagol argued that the first modern feminists in India were Christian women, women such as Laxmibai Tilak and Pandita Ramabai. Using Anagol's definition of feminism as "a theory and practice which presented a challenge to the subordination of women in society and attempted to redress the balance of power between the sexes," this article shows how feminist consciousness was cultivated in Christian schools, churches, hospitals, and organizations in late-nineteenth-century China. Kwok Pui Lan pointed out in 1992 that Christians were the first women in China to band together to fight women's oppression; however, like so many of the claims made about women in global Christianity, this one has not yet been fully appreciated by historians and missiologists. I show how mission schools gave girls access to new models of personhood and womanhood. Likewise, Christian scriptures, churches, and voluntary societies such as the WCTU and YWCA provided space to reflect on gender identity and activism. Through all these avenues, a modest version of Christian feminism was cultivated in China decades before the secular women's movement began in 1900.

**Keywords:** China; feminism; Christianity; women; education

## 1. Introduction

In 2010, Padma Anagol argued that the first modern feminists in India were Christian women, women such as Laxmibai Tilak and Pandita Ramabai (Anagol 2010). These women converted to Christianity from Hinduism at the end of the nineteenth century, and their conversion included a critique of both Hindu caste structure and gender ideology. Anagol noted that Indian feminism "began with a gendered critique of religion through commentaries on religious texts that legitimised patriarchal practices. In what appears to be a global phenomenon, religion has been the principal arena in which women have fashioned their weapons of opposition, providing them with their chief passage to feminist consciousness" (Anagol 2010, p. 525). Building on Anagol's theoretical and historical framework, I argue that just like in India, some Christian women in nineteenth-century China also developed a feminist consciousness. In 1992, Kwok Pui-lan hinted at this argument, but like so many of the claims made about women in global Christianity, this one has not yet been fully appreciated by historians and missiologists (Kwok 1992b, p. 152). Just like in India, Christian women in China used biblical stories of Jesus to critique Confucian gender ideology (Kwok 1992a, pp. 47–51). "Christianity thus provided religious sanctions for women in their struggle against hierarchical social relationships and sexual discrimination. It offered another symbolic universe from which to challenge the 'sacred canopy' that legitimated women's inferior position in Chinese culture" (Kwok 1992a, p. 163).

Most scholars date the beginning of the feminist movement in China to the turn of the twentieth century (Kwok 1992a, p. 108; Edwards 2000, p. 620). At that time, new women's journals began to be published and the women's suffrage movement began in earnest. Scholars like Kwok and Louise Edwards acknowledge that Christian schools and organizations helped to set the stage for the development of feminism in China, but Kwok

has said strongly that "the rising consciousness of Chinese Christian women" came about more through the influence of secular feminists than through the influence of missionaries (Kwok 1992b, p. 152). I am not taking issue with Kwok's basic argument. However, I do want to argue that Christian women were the first groups of women to gather and take action on behalf of other women, which is a point Kwok herself makes. I also want to show how mission schools, hospitals, churches, and organizations such as the Women's Christian Temperance Union (WCTU) and the Young Women's Christian Association (YWCA) created spaces for Chinese women to develop a version of feminist consciousness in the 1870s, 1880s, and 1890s, before the advent of the secular Chinese women's movement.

Because I want to compare the nascent feminist consciousness of Christian women in China with that of women in India, I think it is apropos to use for this paper the definition of feminism that Padma Anagol uses in her paper on India. She defines feminism as "a theory and practice which presented a challenge to the subordination of women in society and attempted to redress the balance of power between the sexes" (Anagol 2010, p. 524). There is evidence that as some Chinese women studied in mission schools, joined anti-footbinding societies, and participated in the work of the WCTU and YWCA, they developed a feminist consciousness. Even though many of these Christian women would not have called themselves feminists, and even though they were not concerned about women's rights in the political and economic realms, they were challenging "the subordination of women in society" in different ways.[1]

The point of drawing attention to feminist consciousness in Christian spaces is not to discredit the role that secular feminism has had in China's history. The point is to show that the history of Christianity does have a feminist strand to it. This strand can often be seen in the first generation of conversions in a given area (Stasson 2021). When looking at the global church today, we can see continued dominance of patriarchy, rampant sex abuse scandals, women lacking access to leadership, and the Bible being used to maintain patriarchy (Du Mez 2020; Johnson and Filemoni-Tofaeono 2003). The reason I want to draw attention to the cultivation of feminist consciousness in both nineteenth-century India and nineteenth-century China is that I want the global church to see that the initial seeds of Christianity that are planted within a culture often have a liberating effect on women. We see this in the early church when we recognize just how radical Paul's letters were in their time. In the Roman world, "any man in a position of power had the right to exploit his inferiors, to use the orifices of a slave or a prostitute to relieve his needs much as he might use a urinal" (Holland 2019, p. 279). Christianity proclaimed that every human person—even slaves and women—had dignity and could not be treated as an object (1 Corinthians 6:19-20, 7:4). "Instincts taken for granted by the Romans [were] recast as sin" (Holland 2019, p. 279). This is the feminist strand in Christianity. In her survey of twentieth-century, Chinese women's autobiographies, Elisabeth Croll found that "in several households women said that they had been attracted to Christianity because of the example it had set of women's equality with and independence from men" (Croll 1995, p. 62). Kwok Pui-lan has said, "Many Christian women believed Christianity to be a motivating force for opposing the oppression of the female sex as well as other forms of social injustice, since Christians believe that human beings are all equal before God" (Kwok 1992a, p. 163).

## 2. Cultivation of Feminist Consciousness within Nineteenth-Century Christian Homes, Churches, and Schools

During the 1920s, Chinese reformers lambasted Christianity for being opposed to the aims of the women's movement, so it might be surprising to see how some nineteenth century, Chinese Christian homes actually provided space for feminist consciousness raising. I found a fascinating example of this in Jennifer Lin's autobiography, *Shanghai Faithful.* She writes about her grandmother, Lin Heping, who dreamed of following in the footsteps of Dr. Xu Jinhong. Dr. Xu had been raised in a Christian home (her dad was "one of the first Methodist pastors in Fuzhou"), and Xu became one of the first Chinese woman to receive missionary funding in order to be trained as a doctor in the United States (Kwok

1992a, p. 117; Robert 1997, p. 181; Ye 1994, p. 319). The year was 1884, and the fact that Xu's parents let her go was completely shocking to traditional Chinese. Dr. Xu returned to China and inspired a generation of women. "Every Christian girl in Fuzhou . . . read about the remarkable achievement of Dr. Xu," writes Jennifer Lin. "The physician Xu Jinhong was a missionary success story, touted from church to church" (Lin 2017, p. 82). Jennifer Lin's grandmother, Lin Heping, was one of those Christian girls who read about Dr. Xu and dreamed of becoming a doctor just like Xu. Lin Heping asked the principle at the Foochow Girls' School, Miss Bonafield, to see if there might be a way for her, too, to travel to the US for medical training. Miss Bonafield told her it was possible if she learned English and performed well in school, so seventeen-year-old Lin convinced her parents (who had converted to Christianity several years earlier) to send her to the more "academically rigorous" McTyeire School for Girls in Shanghai. Another girl with the same dream went with her. They arrived at the school in 1897, a time when "McTyeire was in its infancy with only two dozen students," including Song Qingling (who would become Sun Yat-sen's wife). All seemed to be going in Lin's favor. Her dream of becoming a doctor was coming to fruition. However, then Lin's parents arranged for her to be married.

> The news of their decision, delivered in a letter to McTyeire, shattered Lin Heping. "I knew that useless girls planned to get married, but others could be independent and become teachers, doctors and important people," she wrote many years later. "Me? Finished—go and get married!" (Lin 2017, p. 83)

This quotation from Lin Heping gives a sense of the kind of feminist consciousness that Lin had at age 17, in the year 1897. According to Ryan Dunch, "the first Chinese women to study abroad and to enter professional careers" were Christian women, women such as Dr. Xu (Dunch 2010, p. 327). Dunch has pointed out that "missions were engaged in education for women and girls for a full sixty years before the initial tentative steps toward school-based education for women were taken by the Qing government" (Dunch 2010, p. 328). Even in 1910, when 140,000 girls were attending government schools, some 60,000 still attended Christian schools (Dunch 2009, p. 69). These Christian schools were important places for raising the feminist consciousness of girls such as Lin Heping, as were the homes where these girls were raised. The girls were encouraged to read stories about Christian heroines such as Dr. Xu, and as they read these stories their own desire to go and do likewise was cultivated, even if later their own mothers dashed their plans by arranging for them to be married.

The vast majority of women missionaries in nineteenth-century China—the women who ran the mission schools—were not what we would today call "feminists" (Hunter 1984, p. xv). However, as Shirley Garrett has pointed out, the goals of the women's missionary movement "were defined in the language of feminism" (Garrett 1982, p. 224). Missionaries wanted to see the world's women liberated from injustices such as child marriage, forced prostitution, concubinage, and footbinding (Robert 1997, p. 170). They wanted all women to have access to education and medical care.

The first school for girls in China was established in 1844 by a British missionary, and several American mission schools were opened in the next decade. In the 1840s, it was difficult to convince the Chinese to send their girls to school. Families did not see education as necessary for girls, and upper-class families did not think it proper to allow their daughters to venture outside the family compound (Robert 1997, pp. 171–72). In order to obtain students for their schools, missionaries had to pay parents and adopt orphans (Montgomery 1910, p. 92).

An important boarding school run by three American missionaries was founded in 1858 in Foochow, "the first continuous Methodist girls' school in Asia" (Robert 1997, p. 178). The curriculum included, "Writing and composition, geography, history, arithmetic, astronomy, and useful and ornamental needlework" (Robert 1997, p. 172). Many of the graduates from this school went on to become the first Chinese school teachers, posted at Methodist day schools in various villages.

Another important school was opened by Americans Maria Brown and Mary Porter in 1872. Their school "was the first in China to require the unbinding of feet as a condition of admission" (Robert 1997, p. 174). The practice of women binding their feet with tight cloths to stunt growth began in the tenth century among China's elite, though the ideal of small, dainty feet can be found even earlier in Chinese poetry (Ko 2007, p. 138). "Male desires for bound feet were born of and perpetuated by poetic allusions" (p. 228). By the seventeenth century, footbinding had been thoroughly eroticized and had also become a way in which families in urban China competed with each other to show their wealth and fashion-sense (pp. 128, 193). Between the eighteenth and nineteenth centuries, "footbinding underwent a 180-degree change: from high urban fashion to customary practice expected of average women" (p. 132). As Lily Xiao Hong Lee has pointed out, "A small foot in China, no different from a tiny waist in Victorian England, represented the height of female refinement. For families with marriageable daughters, foot size translated into its own form of currency and a means of achieving upward mobility" (Lee 2016, p. 376).

Missionaries Maria Brown and Mary Porter felt that footbinding did not correspond with what they believed to be God's best for women. They saw footbinding as harmful to women and harmful to society because it kept women from being productive members of society (Dzubinski and Stasson 2021, p. 175). For that reason, they forbid it in the school they established.

Opposition to footbinding spread. According to Kwok Pui-lan, it was this issue that first caused Chinese women to "organize themselves to address the oppression of women." The year was 1874. "Nine working-class, illiterate women formed an antifootbinding society in a church of the London Mission in Xiamen" (Kwok 1992b, p. 152). In the formation of this society—the first of its kind in China—we see the way in which Christian spaces could stimulate nascent feminist consciousness. During the 1870s, these women "continued to hold semi-annual meetings, encouraging members not to bind their daughters' feet, and later the society requested that adult women unbind their feet" (Kwok 1992a, p. 111).

Dana Robert says that "by the 1880s, following the lead of Mary Porter and Maria Brown, progressive Chinese and Chinese Christians increasingly joined women missionaries in efforts to ban social customs detrimental to women" (Robert 1997, p. 175). In addition, Kwok Pui-lan says that "in the 1890s, the [anti-footbinding] movement spread to many cities, supported by girls in mission schools and women in local church groups" (Kwok 1992b, p. 152). During this period, it became common for prominent Chinese women to unbind their feet in public. As they unbound their feet, these women would urge others to do the same (Robert 1997, p. 176).

Dorothy Ko has written about the incongruities in the millennium-long history of footbinding in China. "At once beautiful and ugly, neither voluntary nor coerced, footbinding defies a black-and-white, male-against-female, and good-or-bad way of understanding the world," she argued in *Cinderella's Sisters* (Ko 2007, p. 227). Ko's analysis flies in the face of the dominant narrative handed down by missionaries and Chinese reformers. While missionaries saw Chinese women primarily as victims of patriarchy and social convention, forced to maim their own bodies for the titillation of men, Ko was more focused on the distinctive agency and worldview of premodern Chinese women. They (unlike modern, western missionaries) did not conceive of their body and self as "separate entities." Rather, they viewed their bodies "as both an obstacle and a vehicle to achieving whatever aspirations they may have had" (p. 206). Footbinding caused a woman pain, to be sure, but it was also a way for her to shape her own future. Some women saw footbinding as "an opportunity for social climbing" (pp. 183, 204). "Footbinding was not merely an announcement of status and desirability to the outside world, but also a concrete embodiment of self-respect to the woman herself" (p. 229).

Ko's work is invaluable for helping us see the nuances in footbinding history and for uncovering the way in which the anti-footbinding movements of the 1890s and 1900s were, themselves, harmful to women. These movements were nearly all "male-initiated" (p. 44), and even when they featured a woman onstage publicly unbinding her feet, "she

appeared more as a spectacle" (p. 47) than as a real, complex human being. Male reformers also oversimplified the process of unbinding, depicting it as a one-time thing that would fully liberate the woman concerned. In reality, however, the unbinding process, just like the binding process, was a painful, drawn-out process with varying levels of success (pp. 47–48). Sometimes, "liberated feet" were "more deformed than bound feet" (p. 11).

Brown and Porter could not have foreseen all this when they made unbound feet a prerequisite for attendance at their school; however, their school and other schools where girls chose to unbind ended up becoming places that could support the girls in this painful, unpredictable process. Churches also provided women and girls with the institutional support they needed to undergo the physically and socially painful process of unbinding their feet. As Kwok says, "Many girls needed to put their feet in warm water when the bandages were removed, and some even had to stay in bed for days. Without peer support and personal encouragement from teachers, the girls would have felt very isolated and powerless to face the harsh criticism of relatives and neighbors" (Kwok 1992a, p. 112). In this way, schools and churches provided a space for feminist consciousness to develop and spread, from one student or parishioner to another. As graduates of Christian schools went on to become teachers, they continued to spread the anti-footbinding spirit and provide a supportive environment to their students (Shi 1914, p. 245).

My sense is that the kind of feminist consciousness that developed in mission schools and churches has been overlooked by later scholars because it was so different from the feminist consciousness on display in the first wave of the Chinese women's suffrage movement between the years 1900 and 1913. Suffrage activists produced feminist magazines and strategized on how to win political power. They "engaged in extensive lobbying, protests, and parades." They "stormed the parliamentary chambers" and smashed windows when they were refused a seat at the table of the new Republic of China in 1912 (Edwards 2000, p. 621). This was not the sort of feminist consciousness that was cultivated in mission churches and schools in the 1870s, 1880s, and 1890s. Within these Christian spaces, feminist consciousness developed through reflection on Christian scriptures and Chinese traditions, as well as through relationships with western missionaries and fellow students. Kwok says "missionaries emphasized the compassion of God, used both male and female images of the divine, downplayed the sin of Eve, and stressed that Jesus befriended women" (Kwok 1992b, p. 151). This had the effect of cultivating a feminist consciousness, but it was a consciousness far more subtle and less political than that demonstrated by suffrage activists.

Another reason that scholars have tended to overlook mission schools as a seedbed of feminist consciousness is that missionaries claimed that their primary reason for founding girls' schools was to create "Christian wives and Christian mothers" (Robert 1997, p. 172). However, this more limited vision of education does not counteract the fact that mission schools gave girls access to new models of personhood and womanhood. Liu-Wang Liming, an important Christian advocate of women's suffrage in the early twentieth century, wrote a history of the Chinese Women's Movement. In this book, she highlights the importance of mission schools teaching girls to think. Decades before they would get the vote, girls were liberated by being taught to think, says Liu-Wang (Liu-Wang 1934, p. 79).

During the 1850s and 1860s, the curriculum at mission schools was rather limited, and the aim was mostly to enable the girls to grow in Christian character; however, gradually the curriculum expanded, and as it did so, attendance also increased. In 1872, there were thirty students at the Foochow Girls' School, and a missionary wrote, "There is now no difficulty in procuring just the kind of pupils we desire, and as many as we can accommodate" (Burton 1911, p. 53). In 1887, there were sixty students at Foochow, and missionaries wrote, "We must refuse about twenty applicants this term" (Burton 1911, p. 54). In 1885, at the insistence of Chinese pastors, the Foochow Girls' School had begun to offer courses in English (Burton 1911, pp. 70–71; Robert 1997, p. 179). Other schools followed suit. The McTyeire School for Girls, founded by Southern Methodists in 1892, had a liberal education right from the start (Ross 1996, p. 209). (This was the school that Lin Heping convinced her parents to send her to in 1897 before they pulled her out to get married.)

A survey of five girls' schools in 1900 found that the curricula included "Bible and Christian books, Chinese classics, mathematics, history and geography, and subjects in the sciences such as physics, chemistry, biology, and geology. Music and singing were taught in four schools, and two schools offered English as an optional subject." (Kwok 1992a, p. 107). Expanding to a more liberal curriculum meant that "by the time of its seventieth anniversary, the [Foochow Girls'] school could boast that among its graduates were 11 physicians, 8 kindergartners, 23 preachers' wives, 40 teachers, and 9 sent abroad for advanced study" (Robert 1997, p. 182).

One of the reasons that feminist consciousness was raised in mission boarding schools such as McTyeire and the Foochow Girls' School was because these schools offered an entirely new kind of environment for learning. Girls lived and learned in intimate community with other girls and foreign teachers (and, eventually, Chinese teachers). As Elisabeth Croll says, "The new institutional environment not only served to insulate daughters from the exclusive influence of their families and tradition, but also brought them into contact with a new rhetoric, frequently Western or mission-sponsored, emphasizing female equality . . . At school girls glimpsed an alternative future to the prescribed patterns of the past" (Croll 1995, p. 44). Heidi Ross says of McTyeire, "the school's collective experience offered its students a forum for discussing their personal concerns, as well as untraditional role models in its unmarried, strong-willed, and highly educated teachers" (Ross 1996, p. 216). At McTyeire, every student was expected to cultivate excellence and leadership skills so as to best serve each other, their families, and China. Wong Su-ling, who attended a Christian school in the early twentieth century, described how impressed she was with her teachers:

> How self-assured they were. What dignity and authority they possessed. Just as much as my old [male] teacher. And then the light dawned. Their status was just the same as if they had been men. Here was real equality. This was what I had been feeling the lack of, without quite being able to put it into words for myself. This was the meaning of my growing feeling of injustice over the disabilities imposed by family and clan because I was a girl. I decided then and there that I would be like them. (Wong and Cressy 1952, p. 209)

Wong's account shows that while the mission school did not instigate her feminist consciousness (she talks about having such consciousness earlier in her life), certainly the mission school caused that feminist consciousness to grow. Wong was struck by the fact that her female teachers had a self-assurance, dignity, authority, and status that she was not used to seeing in women. Essentially, the existence and example of her teachers confirmed and added fuel to the fire of her own nascent feminist consciousness.

Wong's account comes from the early twentieth century, but it is possible to find, buried in missionary literature, stories of similar feminist awakenings in earlier generations of students. In the early 1880s, thirty-eight students studied at a girls' boarding school in Kiu Kiang (Twelfth Annual Meeting of the General Executive Committee 1881). A story about one of these students appeared in the pages of the missionary journal *Heathen Woman's Friend.* We are not even told the girl's name, but here is her story, as told by missionary Delia Howe:

> One of our brightest school-girls—a little one, ten or twelve years old—injured her ankle a few days ago quite seriously. It had been pressed so out of its normal shape and size by the binding of the foot that it was impossible to determine the extent of the injury . . . the foot had to be unbound and a normal circulation established as far as possible in the poor deformed half-dead little foot. Some of the school-girls said to the child, "Your mother-in-law will almost kill you for unbinding your foot." The child answered, "I know she will, but I am getting bigger and stronger now, and I can fight her back." (Howe 1881)

Perhaps this girl, like Wong Su-ling, had always had a sense that she wanted her life to be different from the path proscribed for most women in China. Perhaps she had always been self-confident and defiant. Be that as it may, it seems plausible that the mission school

environment cultivated a deeper awareness of her own capabilities. The fact that Delia Howe identified her as "one of our brightest school-girls" suggests that she had received positive attention from her teachers, in acknowledgment of her intellectual prowess. She had hitherto kept her feet bound, but the forced unbinding after her ankle injury seems to have called forth her self-confidence. When classmates told her, "Your mother-in-law will almost kill you for unbinding your foot," the girl replied, "I know she will, but I am getting bigger and stronger now, and I can fight her back." We do not know if these were her exact words—they come to us through Delia Howe, but it is still significant that in the year 1881, Howe chose to include some of her words. Dorothy Ko has pointed out that much of the anti-footbinding activism and literature from the 1890s and 1900s was directed by men. Rarely do we get a glimpse of "the female speaking subject" (Ko 2007, p. 44). In this missionary journal, we are not given the name of the female speaking subject, and we do not see her own penned words, but we do get a sense of the strength of her person, and we do get a sense of her perspective. The student knew her mother-in-law would be furious, but she also recognized her own agency in the relationship: "I can fight her back." Would this 10- or 12-year-old girl have had such confidence if she had never been at this school? We cannot know for sure, but it seems plausible that since this girl had only then chosen to unbind her feet, there was something about the mission school environment that had given her the space to do something that would have been more difficult at home.

Unfortunately, just as public anti-footbinding crusades in the early twentieth-century tended to mute women and treat them as "a spectacle on the podium," (Ko 2007, p. 47), so Delia Howe's commentary also papers over the complex dynamics in the history of footbinding with a simplistic us/them, good/bad dichotomy. At the end of her article, Howe self-righteously exclaims,

> These mothers-in-law would often prefer to let a foot decay and drop off rather than have it unbound, and have their son disgraced by a large-footed woman. This would very probably have been the case with the little girl I told about, had she been under her mother-in-law instead of with us. I do want to help the women here. (Howe 1881)

Missionaries such as Delia Howe were used to seeing the world in terms of a "heathen"/Christian dichotomy (Gin Lum 2022). Heathens were seen as "degraded" and "wretched" (pp. 15–16). The goal for missionaries was to free daughters from such monstrous mothers-in-law, who would rather have "a foot decay and drop off" than "have their son disgraced by a large-footed woman." Kathryn Gin Lum draws attention to the problematic combination of superiority, charity, and desire to control that marks American history. "White Protestant Americans . . . hold themselves to be the heathen's savior . . . The activation of their own humanity and sense of superiority relies on the existence of 'degraded' and 'wretched' heathens for them to feel pity over and save" (Gin Lum 2022, pp. 15–16).

However, despite this important critique of missionaries such as Howe, we still have in this story a girl, ten or twelve years old, determined to *fight back* against footbinding, just as other Chinese women and girls in this period were doing. And it does seem likely that the space of the mission school environment in the early 1880s gave this girl the ability to take actions that would not have been possible in a home environment. The mission school seems to have given her the space to conclude that the right decision at this time was to stop binding her feet, even if her family disapproved.

In 1882, missionary Gertrude Howe (sister of Delia Howe) wrote about two other girls who attended the boarding school in Kiu Kiang. Like the girl described above, they realized, through their time at mission school, that they wanted a different life path from that of a traditional Chinese woman. First, there was Gwi-yin. Gwi-yin's family had given her permission to attend the school for a period of time. However, when she went home to visit, "the family found her too useful to part with her again" (Howe 1882). According to Howe, Gwi-yin was upset. She wanted to be able to finish her studies, but her family would not hear of it. Then, Gwi-yin fell ill. "The Chinese say the illness was induced by

brooding over her troubles," wrote Howe. Tragically, Gwi-yin did not recover. She died in the midst of her troubles, longing for more time at school. Then, Howe told of Chan-sieu. After finishing at school, Chan-sieu returned home. She missed school so much that "she induced her people to allow her to return to the school for the few months" before her ensuing marriage. "Seeing the good which Dr. Bushnell was able to accomplish among the Chinese women, her heart longed to learn something of the practice of medicine, and she urged her people to allow her to remain a few years to this purpose; but they would not hear of it." Chan-sieu was forced to return home, utterly disappointed.

In her article for *Heathen Women's Friend*, Howe concluded with a statement that I would argue hints at the kind of feminist consciousness that was cultivated in mission schools. She wrote, "it might be argued against our school that we raise these girls to *discontent* with the necessary surroundings of their life." Chan-sieu and Gwi-yin had been raised with a certain vision of what constituted womanhood. Whether or not their feminist consciousness had been raised prior to attending school, it seems clear that their brief experience at school nurtured their desire for something different than that which was typically proscribed for Chinese women. In a word, they wanted more education. Chan-sieu dreamed of becoming a doctor like Dr. Bushnell, who was also part of the mission at Kiu-Kiang. Unlike Howe's own adopted daughter, Kang Cheng, Chan-sieu and Gwi-yin were not given the chance to live out their dreams of furthering their education and doing something different with their lives.

A story with a happier ending comes to us from Miss Lucy H. Hoag, who also worked at the boarding school in Kiu Kiang. Hoag tell the story of Tsay-yin, a 17-year-old woman who attended the school. Tsay-yin's husband at first allowed her to be there; however, one day in 1880, he "came and demanded his wife" (Hoag 1881, p. 107). It turned out he had sold her to another man (the two had been having marital troubles, in large part because both of them had a terrible temper). Miss Hoag refused to hand Tsay-yin over to her husband, so he came back drunk and armed with a knife. Eventually, "he was arrested by a native officer," and, once freed, left town. As Miss Hoag told it, "Tsay-yin was only too happy to remain in the school." Tsay-yin also began to help in the mission hospital, but then she got word that "her uncles were trying to sell her" (p. 108). She ran away for a week, and eventually her family gave up on her. She was free to remain unmarried and to work at the hospital again.

The following year, at a woman's meeting, Tsay-yin gave her testimony, which was written up by Dr. Kate Bushnell and printed in *Heathen Woman's Friend*. Tsay-yin's testimony fits the classic evangelical script:

> When I came [to the boarding school] I was exceedingly bad, and with every other bad trait had a fiery temper ... I tried to obey God's commandments and learn to pray. I soon grew to be a little better, but my trouble *always* was my quick temper ... I prayed and prayed for God to forgive the great sin I had committed, and to make my heart pure and clean; and do you know, after a few hours I *just knew* my sins were forgiven! I *knew* I had a clean heart, and Christ dwelt in it, and I have been exceedingly happy ... for now I know that Christ has manifested himself unto me. (Bushnell 1881, pp. 54–55)

Many modern readers will cringe at Tsay-yin's estimation of herself as "exceedingly bad," hearing in this statement the simplistic heathen–Christian paradigm that Tsay-yin likely imbibed from her mission school environment. We do not have enough information to truly understand Tsay-yin's inner consciousness, especially because her words are filtered through those of missionary Kate Bushnell. However, the broad outline of Tsay-yin's life still seems to point to the mission school environment as a place where Tsay-yin could experiment with new forms of womanhood and new forms of personhood. In addition, I argue that we can see Tsay-yin's respect for herself grow. Tsay-yin's husband had mistreated her, and then her own family tried to sell her. Tsay-yin had every reason to simply crumple up and wallow in self-pity. Instead, we find her standing up in a group of women to share about her experience. It is no small thing that Tsay-yin chose to give her



testimony in public like this. Dr. Bushnell, in narrating the story, says that at first Tsay-yin was "bashful" and "flushed" and "clasped her fingers nervously" (p. 54), but then she overcame her fear and launched into her testimony. By the end, "all in the room were visibly affected." On first glance, this might not seem like the kind of nascent feminist consciousness that I am talking about, but I would argue that in Tsay-yin's story, confessing her sin and feeling overcome by a sense of Christ's forgiveness were part of the process whereby she felt empowered and then able to become a medical assistant to Dr. Bushnell.

Mission schools created spaces where girls could envision new dreams for themselves and for China. Because the principals of these schools opposed "bound feet, forced betrothal and the subjection of women in China," and were vocal about their opposition to such things, girls in these schools began to reflect on footbinding and arranged marriage and the various ways that women were disadvantaged in China (Croll 1995, p. 45). In other words, their feminist consciousness was cultivated. Their teachers introduced them to "the ideals of 'freedom', 'individualism', 'self-fulfilment' and 'equality of the sexes'" (p. 46).

Ironically, even though the stated purpose of girls' schools was to creative wives and mothers for Christian men, a large number of the graduates actualized their dreams by staying single (Stasson 2021, p. 180). If their teachers were surprised, they should not have been. That is what they, themselves had done in order to actualize their own dreams as ambitious women from Victorian England and America.[2]

### 3. Cultivation of Feminist Consciousness within Nineteenth-Century Hospitals

Feminist consciousness was not just raised in mission schools, it was also raised through mission hospitals. The Women's Foreign Missionary Society (WFMS) of the Methodist Episcopal Church opened its first hospital for women and children in 1875, and others followed. The social results of these hospitals were tremendous. Not only was women's health improved, but also women gained access to new forms of education and opportunities for work outside the home, as missionary doctors began training them to be nurses and medical assistants (Robert 1997, pp. 165–66). Dana Robert wrote that "although male physicians went out from Europe as early as 1730, their impact as a group did not equal that of women doctors in the late nineteenth century" (Robert 1997, p. 166). I have already mentioned the story of Dr. Xu Jinhong (also known as Hu King-eng, Hu Juying, and Hu Jinying), who went to study at the Woman's Medical College of Philadelphia in 1884. Dr. Xu's study was financed by the WFMS, and she graduated in May of 1894. Even before Dr. Xu, Jin Yunmei (another daughter of a pastor) went to study in America. She studied at the Woman's Medical College in New York and was supported by the Woman's Board of the Dutch Reformed Church. She returned to China in 1888 (Kwok 1992a, p. 117).

Shi Meiyu (known in English as Mary Stone) and Kang Cheng (known as Ida Kahn) were two other Chinese doctors supported by the WFMS, and their story offers further evidence of how feminist consciousness was stimulated through mission churches, schools, and hospitals. Let me begin the story of Shi and Kang by saying a few more things about Methodist missionary Gertrude Howe. Howe arrived in China in 1872 and started the Rulison-Fish Memorial School (the school that Tsay-Yin, Chan-sieu, and Gwi-yin attended). "Opposition to girls' education was so severe that by 1883, what was to become one of the premier girls' college preparatory schools in China still had only ten students" (Robert 1997, p. 185). Over the course of the next decade, however, Howe dedicated herself to giving her students the best education she could. Mr. Shi, pastor of a Protestant church in Kiu Kiang, told Howe that he wanted his daughter (Shi Meiyu) to become a doctor like missionary Kate Bushnell, so in 1892, Howe took "five of her best pupils to Michigan University. She had tutored them in mathematics, chemistry, physics, and Latin so that they could pass the entrance examinations" (Robert 1997, p. 185). Shi Meiyu and Kang Cheng (Howe's adopted daughter) became "the first Asian women to graduate from Michigan in medicine" (Kwok n.d.). In 1896, they went back to China and began establishing dispensaries and hospitals for women and children. Later, they established nursing schools for women. They believed average Chinese women could become empowered agents of

healing. Around 3 in 10 newborns died in China at this point in history, and Shi and Kang trained their nurses to oversee deliveries and improve the odds of infants surviving. Other nursing schools and medical schools were being established at this time, but only upper-class women had the prerequisite education, time, and money to attend. Shi and Kang made it possible for lower-class women to gain a nursing education. This allowed women to avoid being "forced into marriage with undesirable partners or left completely destitute by the death or abandonment of a spouse" (Shemo 2011, p. 89). Ultimately, the example of Shi and Kang "was instrumental in inspiring many Chinese women to become physicians themselves, and in opening the medical profession to Chinese women" (Shemo 2011, p. 1).

In reflecting on the story of Shi and Kang, historian Weili Ye notes, "It was truly remarkable that these women doctors presented themselves in China before the end of the nineteenth century, when few Chinese men had a higher Western education and the majority of Chinese women were not educated at all" (Ye 1994, p. 318). Shi and Kang did not use the term "feminist" for themselves because in their mind that was a term that was used for women with far more radical ideas than their own (Ye 1994, p. 322). However, if we return to Padma Anagol's definition of feminism as "a theory and practice which presented a challenge to the subordination of women in society and attempted to redress the balance of power between the sexes," then certainly Shi and Kang's providing of medical care and medical training for women was a way in which they were challenging the subordination of women to men in China. Why should only men have access to medical care? Why should only men be trained as medical professionals?

In her analysis of Shi and Kang, Weili Ye says "Both Christianity and the medical profession appeared to help the doctors retain their femininity." Shi and Kang were seen as caring for and serving others, which were feminine characteristics. In their vision of womanhood, Shi and Kang "were closer to the missionary female subculture than to the indigenous radical women's movement that emerged after the turn of the century" (Ye 1994, p. 322). This characteristic caused Ye to conclude that "the doctors were not convincing models. To the younger generation of Chinese women who were turning their attention to women's political rights and the issue of gender equality, the doctors' position appeared to be unsatisfactory as meaningful guidance" (p. 324). Even though Shi and Kang looked incredibly conservative when they were compared with secular Chinese feminists in 1910, it is important to remember that back in 1890 they were radical by both Chinese and American standards. Most American medical schools were not open to women in the 1890s. The Journal of the American Medical Association "continued to oppose higher education for women on the grounds that the energy women expended in pursuing professional goals would interfere with their proper reproductive functions" (Women Gaining Access to Medical Education n.d.).

Shi and Kang were elevated by their non-Christian peers as exemplars of China's "new woman." However, these reformers often downplayed or even ignored the fact that Shi and Kang were Christians. They highlighted instead Shi and Kang's education and character and the fact that their medical practice was helping to make China stronger (Ye 1994, p. 320). However, Shi and Kang continued to insist that their Christian faith was not incidental. It was central to their identity, and they wanted it to be part of China's future. Thus, as Connie Shemo says, "Christian evangelism was for Kang [and Shi] a profoundly patriotic activity" (Shemo 2011, pp. 108, 200). These women "didn't see any conflict between their nationalism and their Christianity" (p. 64). What's more, Shi and Kang saw the empowerment of women as an integral part of both their nationalism and their Christianity.

It is important to note that Shi, Kang, and Xu were not the only women doctors in China at the turn of the century. In an article about "What Chinese Women Have Done and Are Doing for China", Shi Meiyu mentions Dr. King of Tientsin, "Dr. Hwang of Shanghai, Dr. Li Bi-chu of Nguchen, [and] Dr. Tsao of Nanking", all of whom were "practicing medicine amongst the women and children" during the first decade of the twentieth century (Shi 1914, p. 243). Margaret Burton mentions Zah Foh-me, who began working

at the Soochow woman's hospital in 1896. A colleague said of her that "for months [she] has borne alone the burden of the hospital" and she "was the first to return to the hospital after the Boxer trouble." For months, she "kept up the work alone and bravely opened the door which we foreigners were not then permitted to enter" (Burton 1911, p. 88). Burton also mentions Miss Yong Hgoh-pau of the Tooker Memorial Hospital in Soochow. Like Shi and Kang, these women were working to expand the access that Chinese women had to medical care.

## 4. Cultivation of Feminist Consciousness within the WCTU

The Woman's Christian Temperance Union (WCTU) was founded by women in the United States in 1874 in order to fight against alcohol consumption. Women were committed to temperance because they saw that male addiction to alcohol led to suffering of wives and children. Under the leadership of Frances Willard, who was president from 1879 until 1898, the WCTU expanded its focus and began to fight prostitution, work for prison reform, and advocate for women's suffrage. The WCTU also expanded its work to regions outside of the United States. In 1884, Willard sent Mary Clement Leavitt on an around-the-world journey to promote the work of the WCTU. WCTU members wrote to missionaries, asking them to assist Leavitt in establishing the WCTU around the world. Leavitt's mission was incredibly successful. In all, "Leavitt logged 97,308 miles, formed 6,623 Unions and talked in 47 languages through 228 interpreters" (Ward 2008, p. 58). She was gone for seven years. She was in China from "October, 1886, to the end of January, 1887," and she formed unions "at Pekin, Tungcho, Tientsin, Shanghai, and Foo-chow." The last dissolved "at once", but she reported in 1891 that "the others still survive, and are doing good work" (Leavitt 1891, p. 9). A union was also established in Chin-kiang. Mrs. Wan served as "President of the Chin-kiang WCTU" from 1889 until 1900 and helped to translate temperance literature (World Woman's Christian Temperance Union 1900, p. 100). Ren Yin S. Mei, who would go on to be a general secretary in the WCTU, wrote that she joined the WCTU when she was just "a little girl," likely with an adult sponsor in 1886 (Ren 1922, p. 49).

In 1893, WCTU missionary Ruth Shaffner reported that the majority of WCTU unions in China were composed of Chinese Christians. Shaffner said that except for the union at Shanghai, which was composed mainly of foreigners, "those secured for the work are mainly from the classes associated in some way with Christian missionaries. Some are from among the native helpers in the hospitals or dispensaries, others are women or girls in the [Christian] schools. Some are teachers, many come from the rank and file of the native Christian church" (Shaffner 1893, p. 200). The WCTU spread information about its meetings through various magazines, including the "two Chinese magazines, *The Illustrated News,* and *Child's Paper*—the publications of the Chinese Religious Tract Society" (Leavitt 1891, p. 65). Shaffner reported that the society in Chin Kiang had twenty-five members and Nanking had forty members; both unions were located at girls' schools. Kiu Kiang had a union of thirty schoolgirls, "who conduct their meetings with no foreign help" (Shaffner 1893, p. 201).

It is true that prior to 1900, the work of the WCTU was still in its infancy in China. Nevertheless, the fact that there were unions—25, 30, and 40 strong—in three girls' schools is still something worth noting. The girls who joined these unions, just like the girls who joined the anti-footbinding societies, were showing that they wanted to gather together to fight against the ills of society, particularly the ills that affected women. At their meetings, they talked about how they could encourage their friends and family members to avoid alcohol and opium. Each girl took a pledge to abstain and, like WCTU women in America, "wore a white ribbon, symbolizing purity" (Kwok 1992a, p. 122). The vision of the World WCTU, of which the unions in China were a part, was to "organize the motherhood of the world for the peace and purity, the protection and exaltation of its homes" (Ward 2008, pp. 57–58). This vision fit well with the mission of girls' schools in China, so housing these unions at girls' schools made a lot of sense. However, as Ian Tyrrell wrote, "The efforts of temperance women to emancipate their sisters from subordination to pre-

vailing customs ironically became enmeshed in the extension of European values and in the domination of large portions of the globe by the imperial powers" (Tyrrell 1991, p. 4). Therefore, just like the rest of the missionary establishment, the WCTU women were open to the critique of imperialism. In the 1890s, the girls who joined the unions at their school basically fell in line with American ideals for the home, but later, as Chinese secretaries took over the national organization from American missionaries, they began to expand the operation of the WCTU to include issues such as "poverty, illiteracy, and the economic dependence of women" (Kwok 1992b, p. 152). Chinese women who became involved with the WCTU (such as Liu-Wang Liming) went on to become leaders in China's suffrage movement (Dubois 2000, p. 547).

## 5. Cultivation of Feminist Consciousness within the YWCA

The Young Women's Christian Association (YWCA) was founded in 1855 by Emma Roberts and Mary Jane Kinnaird in the United Kingdom. It was an organization with a "dual commitment to personal piety and social service" (Boyd 1986, p. 12). Evangelism was important, but so too was providing for the temporal needs of women and girls, who were especially at risk of various injustices brought about by the Industrial Revolution. Organizations of like-minded women also began to form in the United States at this time, such that by the turn of the century, "hundreds of YWCAs were in existence" (Britannica n.d.). These organizations gave girls and women job training, safe housing, and opportunities to meet for prayer or Bible study. They encouraged women to play sports or take classes in nutrition, hygiene, sex education, financial management, or childcare (Boyd 1986, pp. 13–15). As YWCAs started to form in various countries, an international organization—the World YWCA—was founded in London in 1894.

The YWCA was first established in China in 1890, when an American woman visited a girls' school in Hangchow and inspired teachers there to start a branch of the YWCA. The teachers translated the YWCA bylaws into Chinese, and one of the Chinese teachers, Miss Tse Ah Mun, became the first president of the organization. During the school year, Miss Tse met with student members every Sunday. Mrs. Stuart, another teacher, said she considered the YWCA organization to be "one of the most potent factors for good that we have known in the life of our school in developing the Christian life of the girls, in giving them self-reliance, executive ability, and, above all, an interest in the souls of those about them" (Tsai and Haass 1977, pp. 75–76).

Unfortunately, the rest of the story of the YWCA is beyond the scope of this paper because it took place after 1900 and my intention has been to draw attention to the development of feminist consciousness in China prior to the first wave of the women's suffrage movement, which began in 1900. The YWCA went on to become "the largest women's organization in China" (Kwok 1992b, p. 152). YWCA women "initiated projects promoting women's welfare in such areas as adult literacy classes, vocational training, physical education, and overseas scholarship for female students and infant hygiene and nutrition" (Drucker 1979, p. 421). In 1930, Kuei Tsai and Lily Haass produced a study of the history of the YWCA in China and said that the YWCA was once "acclaimed as the forerunner of the women's movement in China" (Drucker 1979, p. 434).

## 6. Conclusions

Feminist consciousness was cultivated in Christian spaces in late-nineteenth-century China. According to Alison Drucker, "It would be a serious error to underestimate the impact upon the early development of the Chinese women's movement of those participants in the American and British women's movements who determined to devote themselves to doing 'women's work' in China, through missionary activities and through quasi-religious organizations [like the YWCA]" (Drucker 1979, p. 424). In this paper, I have attempted to draw attention to their activities. I have shown how the girls who the missionaries taught had their feminist consciousness raised through education, exposure to new models of womanhood, and participation in the YWCA, WCTU, and mission hospitals. These girls

and the women who gathered to oppose footbinding were the earliest contributors to the Chinese women's movement. Drucker defines "women's movement" as "a broad range of efforts united only in their expression of a conscious desire by women to advance the cause of their sex" (Drucker 1979, p. 422). Thus, even though the secular women's movement would overtake the Christian women's movement in number and scope, we should still acknowledge that Chinese Christians were the first women to band together to oppose the subordination of women in Chinese society.

In some ways, this whole paper has been an elaboration of points Kwok Pui-lan has raised. She argued in 1992, "Development of women's consciousness depended on the formation of forums, groups, and networks, in which issues of importance to women could be articulated and publicly addressed. Christian girls' schools, women's groups in churches, local chapters of the WCTU and YWCA provided channels for women to form peer groups and to effect changes in ways acceptable to society" (Kwok 1992a, p. 132). Although Kwok made this argument in 1992, it has not been appreciated sufficiently. In my own article, I have highlighted much of the same history that Kwok described, but I have also added a few more examples of the cultivation of feminist consciousness within girls' schools. I have also added more details about the establishment of the WCTU and YWCA in the 1880s and 1890s. Both organizations became prominent in Chinese society during the early twentieth century, but I wanted to establish that, though small, they did gather girls and women to their cause in the 1890s, thus proving that these organizations did officially precede the first wave of the women's suffrage movement, which Edwards dates as 1900 to 1913.

Both Edwards and Kwok acknowledge the ways in which the Christian schools and organizations set the stage for the secular feminist movement; however, because Christians in China remained dedicated to the idea of women having "distinct roles based on their maternal function," this has been seen in some ways as a belief disqualifying them from being considered "feminists" (Kwok 1992a, p. 133). It is difficult to tell where Kwok falls on this idea, for in some places in her text she does describe "Christian feminists," and she does say that Christian women are one stream in this wider collection of disparate women's movements, a point that Drucker also makes (p. 135). Once again, in this paper, I simply wanted to put an exclamation point on this idea: feminist consciousness was cultivated in Christian spaces prior to 1900. Even if Christians were much more conservative compared to later feminists, even if their idea of women's predominant role in the home is seen as limiting by today's feminists, and even if most of them chose not to call themselves feminists, I still think it is important for the church and academy to see that the spread of Christianity in nineteenth-century China produced feminist consciousness.

**Funding:** This research was funded in part by the Lilly Endowment at Indiana Wesleyan University.

**Acknowledgments:** The author wishes to thank the two reviewers for their helpful reviews of this article.

**Conflicts of Interest:** The author declares no conflict of interest.

## Notes

[1] Webster defines feminism as "belief in and advocacy of the political, economic, and social equality of the sexes expressed especially through organized activity on behalf of women's rights and interests" (https://www.merriam-webster.com/dictionary/feminism (accessed on 6 March 2023). Anagol's definition does not focus primarily on the political realm, nor does it mention "rights."

[2] In 1907, 1038 out of 2481 (about 40%) of women missionaries in China were single (Davin 1992, p. 261).

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
