# Peer review of "Christian Women and the Development of Nascent Feminist Consciousness in Nineteenth-Century China"

_religions, doi:10.3390/rel14030387_

Round 1

Reviewer 1 Report

This paper is an ambitious venture into women's history in China, provoking us to reconsider what exactly is meant by the term "feminism" and who qualifies as "feminist." There are lively anecdotes from missionary journals. 

Some wonderful parts in this paper include: 

- the anecdote about Lin Heping

- lines 231-238, how domestically oriented education was still really significant in giving girls "access to new models of personhood and womanhood." 

- the point that Christianity and the medical profession "appeared to help the doctors retain their femininity" because they were seen as caring and serving others. 

I've struggled to put my finger on this, but I have the overall impression that the eagerness to redefine the term "feminist" or stake a claim to Christian women as "the first feminists in China" creates an argument that is difficult to defend. I wonder if a more modest claim, such as "Christian women and the development of nascent feminist consciousness in China" would be a more defensible argument than "China's First Modern Feminists." Because, instantly, all sorts of terms must be defended and qualified -- "first" "modern" and "feminist". 

I appreciated the direction to Padma Anagol's work and her argument that feminism is defined as resistance to subordination of women to men and that feminist consciousness developed among women in the 19th century, though cloaked in the less contentious language of social reform.

I see the appeal of applying this argument to Chinese Christian women, where similar dynamics were at work. And yet the way the evidence of the paper is structured, currently, the evidence is sometimes uneven. I think it is absolutely true that the Chinese Christian doctors like Shi Meiyu were among China's first feminists if we define feminism as breaking into roles traditionally reserved for men. Is this the same thing as "fighting against subordination to men"? It is absolutely true that women in Christian girls' schools had unprecedented access to educational institutions that came before secular government-run schools. But, again, was going to a Christian girls' school exactly the same thing as "fighting against subordination to men"? 

Feminism is a big topic. It is a big word. It's used for many things. It was not used by the women themselves (the 19th century Chinese women). How important is it to pin this word on them? How important is it to make "redefining the label 'feminism'" the central topic of the paper? Does all the evidence in the paper point toward this redefinition? Is the quality of some 19th century Chinese women's struggle for more autonomy in their bodies, education, or social organizations the same thing as "feminism", and should the paper focus on how this word is being expanded, or is that struggle simply a 19th century Chinese women's struggle for more autonomy in their bodies, education, or social organizations, with a note toward how this was an important precedent of 20th century feminist movements, a proto-feminism? 

Having said all this, I do think it's important that we acknowledge the ways in which religious organizations helped organize women, develop feminist consciousness, and so on. You are absolutely right that we shouldn't disqualify women who are fighting against women's subordination from being partisans in that fight simply because they were religious. Religious, even conservative women can still articulate feminist ideologies. 

Maybe a more productive way to look at the article would be to use "feminist" more as an adjective and less like a noun. Instead of "China's first modern feminists" maybe you could discuss "Chinese Christian women's develop of feminist consciousness" or something like this. 

It just seems like a fraught activity to declare Firsts in Feminists who were Modern. So many terms to define, clarify, disambiguate, connect to existing literature. The current paper lacks this sophisticated connection to those literatures. Some of the examples given don't hold up-- for instance, the various testimonials reported in Heathen Women's Friend, are far too formulaic and processed by external voices (the white missionaries) to give us any real insights into interior feminist consciousness or shifts in consciousness. Sure, this is what the missionaries say is going on in the Chinese women's heads, but how do we know? 

Another gap in the literature is a failure to engage with the white missionaries' entanglement in ideologies dividing people into categories of "heathen" and "Christian." Kathryn Gin Lum's book, Heathen, would be important reading here. The missionaries' posture vis-a-vis the Chinese women is important. For instance, in lines 344-374 there's a discussion of a girl child unbinding her foot. You say "Clearly, this girl had awakened to new potential within herself." What if she was always like that? Where is the evidence of change over time in her inner psychological understandings? How can we know? Is Delia Howe's desire for the child to resist her mother-in-law feminist, or is it resistence to the authority of "heathen" Chinese culture? There are many complicated power dynamics at work and they should be explored, not simplified. 

In sum, there are some wonderful stories and anecdotes here and the overall point, that we should not ignore the agency and organizational capacities and choices of early 19th century CHinese women is well taken. Chinese Christian organizations really should be recognized as nurturing places for the development of feminist consciousness. But the focus on "firsts" and the insistence on using a label that the women weren't using themselves for the women feels a little too insistent, sometimes a little defensive (as in lines 445-447--"Shi and Kang definitely count as two of the first feminists in China"). Why is it so important to use this label? Does it really hold everyone mentioned in this article together? Can we hold them together more precisely? 

In some ways the whole paragraph from 39-51 sounds like the most sensible way to approach this. Yes, the women's journals were being published and the women's suffrage movement began in earnest around the turn of the century. Yes, Christian women were among the first groups of women to gather and take action on behalf of other women, or to develop the women's shared consciousness, before the advent of the secular Chinese women's movement. Is this perhaps enough? Do you have to apply the label and then say, "See, these were the first"? 

Also, the discussion of footbinding seems quite reductionist at times, echoing the strong anti-footbinding rhetoric of the 19th century missionaries. Surely recent works like Dorothy Ko's books and articles on footbinding come into the equation here. As you say in the comparison to a tiny Victorian waist, a small foot meant so many things and carried so much cultural and economic capital. In some ways one might argue that a feminist act was to perform an extremely aggressive binding of a daughter's foot, to allow her to rise in society and experience socioeconomic mobility. 

Author Response

I thoroughly appreciated this reviewer's comments. I am so grateful they took the time to give me such solid feedback. I believe I have addressed all of the issues raised by the reviewer. I read both books recommended to me (Cinderella's Sisters by Ko and Heathen by Gin Lum). Both deepened my understanding tremendously, and I incorporated both into my revised paper. I took the reviewer's advice and chose to rename my paper "Christian Women and the Development of Nascent Feminist Consciousness in China." I got rid of the language about Christian women as the "first feminists" and went with the reviewer's advice in terms of using "feminist" as an adjective rather than a noun. I deleted many parts of the paper and rewrote them. The section I reworked the most can be found between lines 292-362. 

Reviewer 2 Report

The article is clearly written and organized very well. The content is really good and the main thesis is proven throughout these pages.

I learned much not only in terms of various forms of feminism, but also about various approaches that modern Christians have made regarding feminism.

One edit: line 481  addiction

Author Response

Thank you for your encouraging words.